# Peptide-Based Anti-PCSK9 Product for Long-Lasting Management of Hypercholesterolemia

**DOI:** 10.3390/vaccines13090889

**Published:** 2025-08-22

**Authors:** Suresh R. Giri, Akshyaya Chandan Rath, Chitrang J. Trivedi, Bibhuti Bhusan Bhoi, Sandip R. Palode, Vighnesh N. Jadhav, Hitesh Bhayani, Avanishkumar Singh, Chintan Patel, Tushar M. Patel, Niraj M. Sakhrani, Jitendra H. Patel, Niraj A. Shah, Rajendra Chopade, Rajesh Bahekar, Vishwanath Pawar, Rajesh Sundar, Sanjay Bandyopadhyay, Mukul R. Jain

**Affiliations:** 1Department of Pharmacology and Toxicology, Zydus Research Centre, Zydus Lifesciences Ltd., Ahmedabad 382213, India; akshyaya.rath@zyduslife.com (A.C.R.); chitrang.trivedi@zyduslife.com (C.J.T.); bibhuti.bhoi@zyduslife.com (B.B.B.); sr.palode@zyduslife.com (S.R.P.); vighnesh.jadhav@zyduslife.com (V.N.J.); jitendrahpatel@zyduslife.com (J.H.P.); nirajashah@zyduslife.com (N.A.S.); vishwanathpawar@zyduslife.com (V.P.); rajesh.sundar@zyduslife.com (R.S.); 2Department of Cell Biology, Zydus Research Centre, Zydus Lifesciences Ltd., Ahmedabad 382213, India; hitesh.bhayani@zyduslife.com; 3Department of Biotechnology, Zydus Research Centre, Zydus Lifesciences Ltd., Ahmedabad 382213, India; avanishkumarsingh@zyduslife.com (A.S.); chintanpatel@zyduslife.com (C.P.); tusharm.patel@zyduslife.com (T.M.P.); niraj.sakhrani@zyduslife.com (N.M.S.); sanjaybandyopadhyay@zyduslife.com (S.B.); 4Department of Medicinal Chemistry, Zydus Research Centre, Zydus Lifesciences Ltd., Ahmedabad 382213, India; rajendra.chopade@zyduslife.com (R.C.); rajeshbahekar@zyduslife.com (R.B.)

**Keywords:** PCSK9, immunization, peptide-based therapy, hypercholesterolemia

## Abstract

Background/Objectives: Hypercholesterolemia remains a major risk factor for cardiovascular disease and a leading cause of global mortality. Proprotein convertase subtilisin/kexin type 9 (PCSK9) promotes degradation of low-density lipoprotein receptors (LDLR), thereby reducing LDL-cholesterol (LDL-C) clearance. While monoclonal antibodies (mAbs) targeting PCSK9 are effective, their short half-life requires frequent dosing and incurs high treatment costs. This study evaluates a novel peptide-based Anti-PCSK9 product aimed at providing sustained LDL-C reduction. Methods: A novel PCSK9 based-peptide conjugated to diphtheria toxoid (DT) was evaluated in various preclinical models: high-fat diet-fed C57BL/6 mice, APOB100/hCETP transgenic mice, BALB/c mice and normocholesterolemic non-human primates. Immunogenicity (Anti-PCSK9 antibody titers, binding affinity by SPR), pharmacodynamics (LDL-C levels, inhibition of PCSK9-LDLR interaction) and safety were assessed. Toxicity was evaluated in rodents, rabbits and dogs through clinical monitoring, histopathology, organ function and safety pharmacology studies. Results: The Anti-PCSK9 product induced robust and long-lasting immune response in all models antibody titers in BALB/c mice peaked by week 6 and persisted for 12 months. LDL-C reductions of 44% in APOB100/hCETP mice and 37% in C57BL/6 mice correlated with high antibody titers and strong PCSK9-binding affinities (85 and 49 RU), leading to 59% and 58% inhibition of PCSK9-LDLR interaction, respectively. Non-human primates showed sustained responses. No systemic toxicity was observed; injection-site reactions were mild and reversible. No adverse effects were detected on cardiovascular, neurological, or respiratory systems. Conclusions: This peptide-based Anti-PCSK9 therapy offers sustained efficacy and safety, representing a promising long-acting alternative for managing hypercholesterolemia.

## 1. Introduction

Cardiovascular diseases (CVDs), including coronary artery disease, stroke, and heart failure remain the foremost cause of death and disability worldwide. According to the World Health Organization (2023), these conditions account for an estimated 17.9 million deaths yearly, representing nearly one-third of all global deaths. Among the modifiable risk factors contributing to CVD, hypercholesterolemia—particularly elevated levels of low-density lipoprotein cholesterol (LDL-C)—has been firmly established as a major driver of atherosclerosis and related complications [1]. Reducing circulating LDL-C levels has consistently been shown to lower the risk of cardiovascular events, making cholesterol management a cornerstone of preventive cardiology.

In recent decades, the molecular underpinnings of cholesterol homeostasis have been increasingly elucidated, identifying key regulatory pathways and therapeutic targets. One of the most significant discoveries in this area is the role of proprotein convertase subtilisin/kexin type 9 (PCSK9), a liver-derived glycoprotein that modulates cholesterol metabolism by binding to LDL receptors (LDLR) on the surface of hepatocytes. Under normal physiological conditions, LDLR binds to LDL particles and facilitates their internalization and degradation, thereby removing cholesterol from circulation. However, when PCSK9 binds to LDLR, it redirects the receptor toward lysosomal degradation instead of recycling it back to the cell surface. This results in a net reduction of functional LDLR available for LDL clearance, consequently increasing plasma LDL-C levels [2].

Recent advances have significantly expanded our understanding of proprotein convertase subtilisin/kexin type 9 (PCSK9), revealing roles beyond its classical function in hepatic lipid regulation. In addition to promoting the degradation of low-density lipoprotein receptors (LDLR), PCSK9 also interacts with other receptors such as very-low-density lipoprotein receptor (VLDLR), apolipoprotein E receptor 2 (ApoER2), LDL receptor-related protein 1 (LRP1), and CD36. These interactions influence lipid uptake, fatty acid metabolism, and foam cell formation, underscoring PCSK9′s broader involvement in systemic metabolic homeostasis [3]. PCSK9 also plays a pivotal role in inflammation and immune regulation. It activates the NF-κB signaling pathway, leading to the production of pro-inflammatory cytokines, and modulates immune responses by downregulating major histocompatibility complex class I (MHC-I) molecules and upregulating oxidized LDL receptors. These effects link dyslipidemia with vascular inflammation and atherogenesis [3]. Transcriptionally, PCSK9 expression is regulated by sterol regulatory element-binding protein 2 (SREBP-2), which is activated under cholesterol-depleted conditions, forming a feedback loop to maintain cellular cholesterol balance [4].

Importantly, PCSK9 contributes to atherosclerosis by impairing LDL-C metabolism, promoting foam cell formation, and enhancing vascular inflammation, making it a key therapeutic target. The NLRP3 inflammasome, a central mediator of vascular inflammation, is closely linked to PCSK9 in a positive feedback loop that amplifies lipid accumulation and inflammatory responses. NLRP3 signaling enhances PCSK9 secretion via IL-1β and MAPK pathways, while PCSK9 in turn stimulates cytokine production in vascular cells and macrophages [5]. Evidence from preclinical studies supports the therapeutic potential of PCSK9 inhibition. For example, immunization with the peptide vaccine AT04A reduced vascular inflammation and necrotic core formation without affecting macrophage numbers—likely by preventing lipid overload and suppressing NLRP3 activation [6]. Similarly, we believe that our Anti-PCSK9 product, by inhibiting PCSK9 and lowering LDL-C levels, may also reduce foam cell formation and suppress NF-κB signaling and inflammasome activation in macrophages, thereby limiting IL-1β and caspase-1 activity. Disruption of the PCSK9–NLRP3 inflammatory loop represents a promising dual-action strategy to stabilize atherosclerotic plaques, reduce vascular injury, and improve cardiovascular outcomes [5]. These mechanistic insights are supported by clinical data from therapies such as inclisiran, an siRNA, and evolocumab, a monoclonal antibody targeting PCSK9. Both have demonstrated significant reductions in LDL-C levels and cardiovascular events in patients with atherosclerotic cardiovascular disease (ASCVD), as shown in the ORION-10, ORION-11, and FOURIER trials [7,8]

Targeting PCSK9 has emerged as a highly effective LDL-C reduction and CVD prevention strategy. Monoclonal antibodies (mAbs) such as alirocumab and evolocumab have been developed to block the interaction between PCSK9 and LDLR, preserving receptor activity and promoting LDL clearance. Clinical trials have demonstrated that these therapies can reduce LDL-C levels by up to 60% when administered in combination with statins and importantly, they have also shown a significant reduction in major adverse cardiovascular events [8,9]. Despite their efficacy, these mAbs come with notable limitations. These biological products require a subcutaneous injection every two to four weeks, which may affect patient compliance. Moreover, the high cost of these agents limits accessibility, particularly in resource-constrained healthcare systems. Long-term use may also raise concerns about immunogenicity and the development of neutralizing antibodies [8].

Alternative approaches have been explored to address these limitations. Small interfering RNA (siRNA) therapies such as Inclisiran inhibit PCSK9 synthesis at the mRNA level, offering longer dosing intervals—typically twice yearly after the initial loading dose. While this approach reduces injection frequency and offers improved convenience, it remains expensive and requires healthcare provider administration, which can pose logistical challenges for widespread implementation [8,10]. Consequently, there is a growing demand for cost-effective, long-acting therapies that combine efficacy, safety, and ease of administration.

One such promising approach being explored is peptide-based immunotherapy, which represents a novel paradigm in PCSK9 inhibition. Unlike passive immunization strategies that rely on exogenous antibodies or synthetic gene silencing agents, peptide-based Anti-PCSK9 products stimulate the host’s immune system to generate endogenous antibodies against PCSK9. These antibodies can neutralize circulating PCSK9, prevent LDLR degradation and enhance LDL-C clearance from the bloodstream [6,8,11]. This approach offers several theoretical advantages: the potential for long-lasting effects from a single or multi-dose administration, reduced manufacturing complexity and improved affordability. Furthermore, peptide-based strategies may elicit a more durable immune response with minimal need for repeated dosing, significantly enhancing treatment adherence and clinical outcomes [6].

In this report, we are describing a novel peptide-based Anti-PCSK9 product and its profile in a range of preclinical animal models.

## 2. Materials and Methods

### 2.1. Anti-PCSK9 Product Formulation

The Anti-PCSK9 product consists of 12 amino acids PCSK9-peptide, a combination of amino acids of PCSK9 protein from regions (aa153–163), which are further modified with unnatural amino acids. This was synthesized using solid phase synthesis and characterized using various analytical techniques at Zydus Research Centre. This PCSK9-peptide was further conjugated with an immunogenic carrier, diphtheria toxoid (DT, supplied by Zydus Vaccine Technology Centre, Ahmedabad, Gujarat). Peptide-DT conjugate was prepared by first desalting DT using Econo-Pac 10 DG columns and quantifying protein concentration via the BCA assay. DT was then reacted with SMPH (succinimidyl 6-[(beta-maleimidopropionamido) hexanoate]), (Sigma-Aldrich, Cat. No. 803626) to introduce maleimide groups. The peptide solution was added to the activated DT and the conjugation was allowed to proceed for 3 h at room temperature, followed by desalting and filtration (Figure 1). The extent of conjugation was evaluated using a citrulline estimation assay, exploiting the presence of a single citrulline residue per peptide to calculate the Peptide:DT molar ratio. The conjugate was mixed with alum adjuvant for vaccine formulation and incubated overnight at 2–8 °C. Monophosphoryl lipid A (MPLA-SM Vaccigrade, Invivogen, Cat. code vac-mpla) was then added and the final formulation was prepared by adjusting with PBS and incubating for 1 h before use in animal studies.

### 2.2. Animals and Immunization

Mice bred at the Zydus Research Centre (ZRC) were housed in individually ventilated cages under controlled temperature (25 ± 3 °C) and humidity (50–70%) with a 12 h light/dark cycle. Animals had free access to a standard rodent diet (Teklad 2018C, Harlan Laboratories, USA) and water. All animal procedures were conducted at AAALAC-accredited facility at Zydus Research Centre following approval by the Institutional Animal Ethics Committee (IAEC) of Zydus Lifesciences Limited. Large animal protocols were also approved by CCSEA, Govt. of India. Female BALB/c mice (6–7 weeks old) (doses 5 µg, 25 µg, 50 µg, 100 µg, and 200 µg/mice), 10 week high-fat diet (60% kcal fat, D-12492, Research diet Inc.) fed male C57BL/6 mice (6–7 weeks old) (doses 25 µg, 50 µg/mice), female hCETP/APOB100 double transgenic mice (18–31 weeks old) (doses 25 µg, 50 µg/mice) and normocholesterolemic female rhesus monkeys (>3 years old, 4–9 kg) (dose 50 µg/animal) were used in the study. Animals were randomized into groups based on baseline serum total cholesterol, LDL cholesterol and body weight. Immunizations were administered subcutaneously with 0.5 mL of the Anti-PCSK9 formulation on Day 1, followed by booster doses at Weeks 2 and 4. Blood samples were collected prior to immunization and at designated intervals up to 6 or 12 months. Serum biochemistry was performed to measure total cholesterol and LDL cholesterol levels using an autoanalyzer (Mindray BS240 biochemical autoanalyzer). Anti-PCSK9 antibody titers were quantified using an indirect ELISA. The binding affinity of these antibodies to PCSK9 was confirmed using a Surface Plasmon Resonance (SPR) assay. The functional activity of the antibodies was assessed through inhibition of the PCSK9-LDLR interaction using a modified ELISA assay. Body weights and clinical signs were monitored throughout the study period to evaluate safety and efficacy.

### 2.3. Safety Profile Studies

Safety pharmacology studies were conducted to evaluate the potential safety concerns of the Anti-PCSK9 product administered subcutaneously at a dose of 50 µg biweekly for neurobehavioral evaluation in Wistar rat. Safety pharmacology studies for assessment of effects on central nervous, cardiovascular, and respiratory systems were conducted following single dose of the product. Neurobehavioral assessments were performed using the Irwin test in male and female rats from the negative control, placebo, and 50 µg dose groups at the end of the dosing and recovery periods. Effect on motor coordination was assessed in ICR mice using the rotarod test, with the latency to fall recorded at 30 and 60 min post-administration of either placebo or the 50 µg Anti-PCSK9 product.

Cardiovascular safety was evaluated in telemetered beagle dogs administered either a placebo or 50 µg of the Anti-PCSK9 product. In this study effects on blood pressure, heart rate, ECG, and core body temperature were monitored before and up to 24 h post-dosing. Baseline values were established from pre-dose measurements and delta values were analyzed using two-way ANOVA followed by Dunnett’s test.

Respiratory function was assessed in conscious male Sprague Dawley rats using a whole-body plethysmograph to measure tidal volume, respiratory rate, and minute volume. Following acclimatization and habituation to the plethysmograph chamber, rats received either a placebo or 50 µg of the Anti-PCSK9 product via the subcutaneous route. Respiratory signals were recorded before dosing and from 85 to 120 min post-dosing. Delta values and area under the curve (AUC) were calculated to evaluate treatment effects. All statistical analyses were performed using SAS^®^ version 9.4.

### 2.4. Toxicity Studies

#### 2.4.1. Acute Toxicity Study of Anti-PCSK9 Product

Local tolerance and general toxicological assessments of the Anti-PCSK9 product were conducted in ICR mice and Wistar rats following subcutaneous administration. In the mouse study, a group of 10 ICR mice (5 per sex) received 100 µg/animal of the Anti-PCSK9 formulation via subcutaneous injection at two dorsal flank sites (0.5 mL per site; total volume, 1 mL). A concurrent vehicle control group (5 per sex) received the placebo. In a separate study, 10 Wistar rats (5 per sex) were administered 200 µg/animal of the Anti-PCSK9 product subcutaneously at three sites with a total volume of 2 mL, while the control group (5 per sex) received the placebo.

Parameters evaluated in both studies included morbidity and mortality, daily clinical signs, local tolerance at the injection sites, body weights on Days 1, 3, 7, and 14, detailed clinical examinations, gross pathology, and microscopic examination of tissues exhibiting gross lesions.

#### 2.4.2. Repeat-Dose Toxicity Study of the Anti-PCSK9 Product

A repeat-dose toxicity study was conducted in New Zealand White rabbits to evaluate the safety profile of the Anti-PCSK9 product. The study included five main and two recovery groups, each consisting of three rabbits per sex. The test item was administered subcutaneously at 100 µg/animal doses on Days 1, 15, and 29. The total dose volume was 1 mL per animal, divided between two injection sites (0.5 mL/site). Control groups included a vehicle control (placebo) and a negative control receiving normal saline to distinguish any placebo-related effects. To evaluate the potential for reversibility, persistence, or delayed onset of adverse effects, recovery groups (three animals per sex per group) were observed for an additional 2-week post-treatment period. Study parameters included morbidity and mortality, clinical signs, detailed clinical and ophthalmic examinations, body weight, feed consumption, clinical pathology, organ weights, immunogenicity, and gross histopathological examinations.

### 2.5. Evaluation of Antibody Titre Against PCSK9

At designated intervals, serum samples were collected from all animal groups. Ninety-six-well ELISA plates (#442402) were pre-coated with 100 ng/well of PCSK9 protein [PCSK9-His tag; BPS Bioscience, Cat. No. 71204] and incubated overnight at 2–8 °C. The following day, unbound protein was removed by washing and wells were blocked with 5% skimmed milk in PBS (Sigma, Cat. No. D5652-10 × 1 L) for one hour. Plates were washed with PBST (0.25% *v/v* Tween-20 in PBS).

Serum samples were serially diluted (1:100 to 1:100,000) using 0.25% skimmed milk in PBS. Diluted samples and controls were added to the wells and incubated at 37 °C for one hour. Subsequently, horse radish peroxidase (HRP)-conjugated anti-mouse or anti-monkey IgG (depending on the serum species) was added and incubated at 37 °C for one hour. After washing, Tetramethylbenzidine (TMB) substrate solution was added and the reaction was incubated for 7 min at room temperature in the dark. Absorbance was measured at 450 nm using a Biotek Synergy HT multidetection microplate reader. Using GraphPad Prism software 8.0.1, titers were calculated using the endpoint dilution method with an OD of 0.5 as the cut-off.

### 2.6. LDLR–PCSK9 Interaction Inhibition Assay

The inhibition of LDLR–PCSK9 interaction by antibodies in serum from Anti-PCSK9-treated animals was evaluated using the BPS Bioscience in vitro PCSK9 [Biotinylated]-LDLR Binding Assay Kit (Cat. No. 72002, BPS Bioscience, San Diego, CA, USA), with minor modifications. LDLR ectodomain (50 µL/well at 2 µg/mL) was coated onto 96-well plates and incubated overnight. On the following day, PCSK9-biotin was diluted in assay buffer to 2.5 ng/µL (50 ng/20 µL) and a master mix containing the diluted serum (test inhibitor) and buffer was added to the LDLR-coated plate. The plate was incubated at room temperature for 2 h.

Streptavidin-HRP was then added, followed by HRP substrate for chemiluminescent detection. Chemiluminescence intensity was measured using a Biotek Synergy HT multidetection microplate reader microplate reader. Relative inhibition was calculated as the percentage reduction in PCSK9–LDLR binding in Anti-PCSK9-treated samples compared to placebo-treated controls (set as 100%).

### 2.7. Affinity Determination by Surface Plasmon Resonance (SPR)

The binding affinity of Anti-PCSK9-induced antibodies in serum samples was determined using surface plasmon resonance (SPR) at 25 °C with a Biacore T200 system. Recombinant human PCSK9 protein was immobilized on a CM5 sensor chip using amine-coupling chemistry; one flow cell was left blank for reference subtraction. Serum samples were diluted 1:10 in running buffer (10 mM HEPES, 250 mM NaCl, 3 mM EDTA, 0.05% *v/v* Surfactant P20) injected over both the blank and ligand-coated flow cells. Each cycle consisted of a 60-s association phase and a 120-s dissociation phase. The needle wash was performed with 40 mM NaOH and 20% 2-propanol, followed by system regeneration with glycine/HCl (pH 1.5) and NaOH washes. SPR data were analyzed using Biacore T200 Evaluation software. Binding curves were generated after reference subtraction and baselines were adjusted to zero. Results were expressed as stability response units (RU), averaged over a 5-s window taken 10 s post-injection. This method enabled accurate characterization of antibody binding affinity and stability in treated serum samples.

## 3. Results

### 3.1. Immunogenicity and Therapeutic Evaluation of Anti-PCSK9 Product in BALB/c Mice

The ELISA antibody titers showed a clear dose-dependent response, with higher doses inducing stronger and more sustained antibody levels. Peak titers were observed at Week 6: 9953 (5 µg), 15,550 (25 µg), 19,259 (50 µg), 24,262 (100 µg) and 25,602 (200 µg), followed by a gradual decline over 12 months (Figure 1a). Functionally, LDLR-PCSK9 inhibition peaked between 6 weeks and 2 months (64–76%) and gradually decreased over 12 months (26–34%). However, a sustained inhibitory effect was observed across all doses, even at later time points (Figure 1b). SPR binding data demonstrated a dose-dependent increase, with the 200 µg group showing the highest binding affinity at Week 6 (31 RU), indicating robust target engagement

These findings highlight the superior efficacy of 25–200 µg doses across all parameters, justifying the selection of 25 µg and 50 µg doses for further studies in various animal models.

### 3.2. Immunogenicity and Lipid-Lowering Effects of Anti-PCSK9 Product in Diet-Induced Hypercholesterolemic C57BL/6 Mice

Immunization with the Anti-PCSK9 product (25 µg and 50 µg per mouse) induced robust immunogenicity and significant lipid-lowering in diet-induced hypercholesterolemic C57BL/6 mice. Serum LDL-C levels declined by 28% (25 µg) and 35% (50 µg) at 3 months and remained reduced at 6 months (33% and 29%, respectively) (Figure 2a). Total cholesterol showed maximal reductions at 3 months (17% and 16%), sustained through 6 months (Figure 2b).

ELISA antibody titers peaked at 2 months—1524 (25 µg) and 3113 (50 µg)—and remained elevated at 6 months (1282 and 2092, respectively) (Figure 2c). Functional inhibition of PCSK9-LDLR interaction was most significant at 2 months: 44% (25 µg) and 58% (50 µg) (Table 1). SPR binding confirmed strong PCSK9 engagement: 31 RU (25 µg) and 49 RU (50 µg).

These results underscore the therapeutic potential of the 25 µg and 50 µg doses for sustained lipid control and immune response.

### 3.3. Efficacy and Immunogenicity in hCETP/APOB100 Double Transgenic Mice

The Anti-PCSK9 product was further evaluated in hCETP/APOB100 double transgenic mice using 25 µg and 50 µg doses over 6 months. Both doses significantly reduced serum LDL cholesterol (LDL-C) and total cholesterol levels. The 25-µg dose showed a maximal 30% reduction at Week 6, while the 50-µg dose reduced LDL-C levels by 44% (Figure 3a). These reductions in LDL-C were sustained at 6 months (24% and 25%, respectively). Total cholesterol levels also declined substantially: the 50 µg group peaked at 33% reduction (Week 6), while the 25 µg group showed 25% (Figure 3b). At 6 months, reductions in total cholesterol remained 14% and 12%, respectively. The high levels of antibody in ELISA confirmed robust immune responses. Peak titers at 2 months were 10,791 (25 µg) and 17,906 (50 µg), remaining elevated at 6 months (927 and 1317, respectively, Figure 3c). SPR assay showed significant binding of these antibodies to PCSK9, at Week 6 showed dose-dependent 59 RU (25 µg) and 85 RU (50 µg), compared to negligible placebo binding (Table 2). LDLR-PCSK9 binding inhibition assays revealed potent inhibition at Week 6: 54% (25 µg) and 59% (50 µg), confirming early and potent PCSK9 blockade.

### 3.4. Immunogenicity Evaluation in Non-Human Primates (NHPs)

The immunogenicity of the Anti-PCSK9 product (50 µg/0.5 mL/animal, s.c.) was assessed over 6 months in normocholesterolemic non-human primates. Though lipid reduction was minimal, antibody titers peaked at 3 months (10,653), significantly above placebo. Titers declined but remained elevated through 6 months (Figure 4a). LDLR-PCSK9 interaction inhibition was evident by Week 6 (31%), peaking at 3 months (50%) and declining to 31% by 6 months (Figure 4b). SPR binding assays confirmed significant target engagement up to 3 months (Figure 4c). These results demonstrate robust immunogenicity and biological activity in the NHP model.

### 3.5. Safety Studies

Safety pharmacology studies evaluated the Anti-PCSK9 product’s impact on the central nervous, cardiovascular, and respiratory systems. Subcutaneous administration of 50 µg once or biweekly administration showed no adverse effects on motor coordination in ICR mice nor neurobehavior in Wistar rats.

In beagle dogs, cardiovascular assessments revealed no changes in ECG, heart rate, blood pressure, or body temperature. No significant respiratory effects were observed using whole-body plethysmography in male Sprague Dawley rats. These results support the product’s safety profile (Table 3).

### 3.6. Toxicity Studies

#### 3.6.1. Acute Toxicity

Single-dose toxicity studies in ICR mice and Wistar rats (100 µg and 200 µg, respectively, s.c.) showed the product was well tolerated, with no systemic toxicity or gross pathological findings. Localized mild to moderate injection site inflammation was observed, consistent with adjuvants (MPL-A and alum).

#### 3.6.2. Repeated Dose Toxicity in New Zealand White Rabbits

Rabbits received 100 µg s.c. dose on Day 1, 15, and 29. No systemic toxicity, mortality, or adverse effects were observed. Localized nodules were noted, with histopathology revealing only mild inflammatory changes and reactive lymph nodes. These effects were non-adverse and reversible. The no observed adverse effect level (NOAEL) was determined to be 100 µg/animal.

## 4. Discussion

The present study demonstrates the robust immunogenicity, therapeutic efficacy, and safety of a novel Anti-PCSK9 product across multiple preclinical models. The product elicited dose-dependent Anti-PCSK9 antibody titers, potent functional inhibition of the PCSK9–LDLR interaction, and sustained lipid-lowering effects, supporting its potential as a long-acting and cost-effective therapeutic alternative to monoclonal antibodies for hypercholesterolemia.

During the last decade, PCSK9 inhibition has emerged as a critical target in lipid regulation. PCSK9, by binding to LDL receptors (LDLR) and promoting their lysosomal degradation, impairs cholesterol clearance from plasma. Inhibition of PCSK9 preserves LDLR function and enhances LDL-C removal, and this mechanism underlines the efficacy of approved PCSK9-targeting monoclonal antibodies (mAbs) such as alirocumab and evolocumab [9,10]. These agents have demonstrated up to 60% LDL-C reductions and a concomitant decrease in major cardiovascular events in large clinical trials [10]. However, their widespread use is limited by several practical challenges. These include the need for subcutaneous injections every 2 to 4 weeks, which may reduce long-term adherence, the high annual treatment cost (USD 4500–6000 per patient), and concerns about immunogenicity with long-term administration [12,13,14].

In contrast, the peptide-based therapy evaluated in this study represents a cost-effective and patient-friendly alternative. The approach adopted in this work relies on active immunization, stimulating the host to generate endogenous antibodies against PCSK9, thereby offering long-lasting cholesterol-lowering effects with limited dosing. The immunogenicity data across multiple animal models—including BALB/c mice, diet-induced hypercholesterolemic C57BL/6 mice, hCETP/APOB100 transgenic mice, and non-human primates (NHPs)—demonstrated a clear dose–response relationship with high antibody titers persisting for up to 6 to 12 months. The antibodies effectively inhibited PCSK9-LDLR interactions, as confirmed by ELISA-based assays and SPR binding, indicating robust and functional immune engagement with the target. The lipid-lowering efficacy was consistently observed in physiologically relevant models. In C57BL/6 mice fed a high-fat diet, LDL-C levels were reduced by up to 35% after immunization, with sustained effects over 6 months. Similarly, in hCETP/APOB100 mice—widely regarded as a more translatable model of human lipid metabolism—LDL-C reductions reached 44% and total cholesterol declined by up to 33% at peak. These results suggest that this Anti-PCSK9 peptide therapy may offer comparable efficacy to mAbs but with a significantly simplified dosing schedule, likely requiring only two to three doses per year [15,16]. Immunogenicity in non-human primates further supports translational potential. Though the NHPs were normocholesterolemic and did not show lipid reduction, the immune responses were significant, with antibody titers peaking at three months and sustained LDLR-PCSK9 interaction inhibition observed through six months. These findings mirror those seen in rodent models and suggest feasibility for future translation into human trials. Studies in NHPs have historically provided reliable indicators for human immunogenicity and vaccine performance [17]. Importantly, the safety profile of the Anti-PCSK9 vaccine was favorable across all preclinical studies. Acute and repeated-dose toxicity studies revealed no systemic toxicities or organ pathologies. Safety pharmacology evaluations in rodents and dogs showed no adverse effects on neurological, cardiovascular, or respiratory function. Injection site reactions were mild and reversible, consistent with the use of adjuvants such as MPL-A and alum. Histological analysis in rabbits confirmed only mild, transient inflammation. The NOAEL was established at 100 µg/animal, which supports the tolerability of clinical dose equivalents. These results are particularly encouraging, as safety is a major requirement for therapies aimed at long-term prevention in asymptomatic patients [18,19,20].

Our findings are in line with previous studies describing similar approaches, commonly referred to as “PCSK9 vaccines”, which aim to elicit active immunity against proprotein convertase subtilisin/kexin type 9 (PCSK9) to achieve long-lasting lipid-lowering effects. Various vaccine platforms targeting PCSK9 have been developed, including peptide conjugates, virus-like particles (VLPs), and nanoparticle-based formulations, with differing degrees of success demonstrated in both preclinical and early clinical evaluations [6,21,22,23] Among the most extensively studied is AFFITOPE^®^ AT04A, developed by AFFiRiS AG, which utilizes a short PCSK9-mimetic peptide conjugated to the carrier protein Keyhole Limpet Hemocyanin (KLH) and adjuvanted with aluminum hydroxide. In preclinical studies, AT04A demonstrated approximately 50–55% LDL-C reduction in Ldlr^+^/^−^ mice [23] while in a Phase I clinical trial, it achieved a more modest LDL-C reduction of around 11%, with a favorable safety profile [6]. While it has promising effects in preclinical models, the use of KLH as a carrier presents certain drawbacks. KLH is not approved as a therapeutic component in any licensed human vaccine and is predominantly used in experimental settings [24]. Its high immunogenicity can lead to a dominant anti-KLH antibody response, which may diminish the immune response to the intended antigen (i.e., the PCSK9 peptide), particularly upon repeated administration [25]. Pre-existing anti-KLH antibodies can accelerate clearance of the conjugate, reducing its efficacy. Additionally, KLH’s marine origin and structural heterogeneity present manufacturing and regulatory challenges, especially in ensuring consistency and scalability [24]. Furthermore, the modest LDL-C reductions observed in the Phase I study (ranging from 7–13%) are notably lower than those typically achieved with PCSK9 monoclonal antibodies, which reduce LDL-C by approximately 60%, highlighting the need for further optimization of such vaccine platforms [25].

From a public health and manufacturing standpoint, peptide-based therapies offer several additional advantages over biologics. Scalable solid-phase methods can synthesize peptides, do not require complex mammalian cell culture systems, and are faster, scalable, and more cost-effective in production, making them significantly less expensive and more accessible globally [18,19]. In resource-limited settings, where cardiovascular disease is rising sharply, such an approach could greatly enhance equity in treatment availability and adherence [14]. In addition, this strategy opens avenues for future multi-target peptide-based therapies that combine Anti-PCSK9 peptides with other atheroprotective targets, potentially broadening the scope of cardiovascular disease prevention. The durability of the immune response, with limited dosing requirements, may improve adherence and long-term outcomes more effectively than daily pills or monthly injections, especially in patients with low health literacy or poor access to healthcare facilities [26].

In conclusion, this study establishes a novel and promising peptide-based product targeting PCSK9, offering durable immunogenicity, functional inhibition of PCSK9-LDLR binding, sustained lipid-lowering efficacy, and a favorable safety profile. Given its lower complexity, improved patient convenience, and potential affordability, this product represents a transformative step toward accessible, long-acting therapies for hypercholesterolemia and cardiovascular risk reduction. Clinical development is warranted to evaluate safety, immunogenicity, and LDL-C-lowering efficacy in human populations.

## 5. Patents

Jain, M., Giri, S., Bahekar, R., Gupta, G., Chopade, R. (2018): Novel peptide-based PCSK9 vaccine. WIPO patent: WO 2018/189705 A1 [27].

## Data Availability

The authors declare that all the data supporting the findings of this study are available within the paper.

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
