# Peer review of "Peptide-Based Anti-PCSK9 Product for Long-Lasting Management of Hypercholesterolemia"

_vaccines, 2025, doi:10.3390/vaccines13090889_

Round 1
Reviewer 1 Report
Comments and Suggestions for Authors
In this interesting work, the authors focus on the study of a PCSK9 peptide conjugated to diphtheria toxoid (DT), applying it to healthy animal models, or those subjected to a high-fat diet or transgenic animals. They studied both pharmacological and immunological aspects with excellent results, particularly in murine models, which demonstrated a very good and long-lasting immune response. High antibody titers were accompanied by a reduction in lipid markers, such as LDL, especially in mice. Points: 1- Endothelial aspect: Can the use of this new PCSK9 antipeptide conjugated to diphtheria toxoid somehow exert an effect on the endothelial level? Could it have repercussions on the ratio of vascular platelets, coagulation factors, inflammatory cells, and endothelial cells during coagulation and hemostasis? 2- Given the close relationship between PCSK9 and NLRP3 inflammasomes (cite and comment on PMID: 36911736, PMID: 40227195), do the authors have any data/hypotheses regarding the use of their innovative PCSK9 antipeptide conjugated to diphtheria toxoid in this relationship?
Reviewer 2 Report
Comments and Suggestions for Authors
The submitted manuscript presents a novel and well-executed preclinical study evaluating a peptide-based active immunization strategy targeting PCSK9 as a long-acting alternative for LDL-cholesterol reduction. The concept is clinically relevant and addresses major limitations of existing therapies such as monoclonal antibodies and siRNA-based inhibitors, including dosing frequency, manufacturing cost, and accessibility.
The introduction is generally well structured, but would benefit from further elaboration. In particular, it would strengthen the rationale of the study to provide a brief overview of the key signaling pathways involving PCSK9, including its role in lipid metabolism beyond hepatic LDL receptor degradation. The introduction should also include a concise summary of the broader landscape of PCSK9-targeted interventions (monoclonal antibodies, siRNA, vaccines, etc.), emphasizing where the present approach fits in.
The results are clearly presented and appropriately analyzed. The immunogenicity, pharmacodynamic effects, and safety data are robust and well visualized. The use of multiple animal models (BALB/c, C57BL/6, hCETP/APOB100, and non-human primates) enhances the translational potential of the findings. The discussion is well written, but could be improved by expanding on the topic of immunogenicity—not only as the mechanism of action, but also as a potential safety concern, especially considering the induction of endogenous antibodies against a native protein. In this context, the authors may briefly comment on the risk of neutralizing or autoreactive antibodies and the implications for long-term tolerability and retreatment.
It would also be valuable to comment on how the observed antibody titers and pharmacodynamic effects in animal models (particularly in NHPs) might translate to humans, including any anticipated differences in dose-response relationships or immunogenicity kinetics.
Overall, this is an interesting and timely manuscript. Following minor revision with expanded discussion on PCSK9 biology, immunogenicity, and translational considerations, the paper will be suitable for publication.
Reviewer 3 Report
Comments and Suggestions for Authors
- While preclinical results are robust, no Phase I clinical trial data is included to validate safety/efficacy in humans.
- Modest LDL-C Reduction in NHPs – Only 31–50% inhibition observed, weaker than in rodents. Could raise concerns about translational efficacy.
- KLH vs. Diphtheria Toxoid (DT) Carrier – Claims superiority over KLH-based vaccines (e.g., AT04A) but lacks direct comparison data.
- Injection Site Reactions – Mild inflammation noted, but long-term effects of adjuvants (MPL-A/alum) need further study.
- No Comparison to Existing Therapies – Does not benchmark against inclisiran (siRNA) or mAbs in the same models.
- Include a Schematic Diagram – Illustrate the peptide-DT conjugate design, mechanism of action, and key findings.
- Compare with KLH-Based Vaccines – Add data on why DT is superior (e.g., immunogenicity, manufacturing ease).
- Expand on SPR Results – Provide kinetic parameters (KD, Kon/Koff) for antibody-PCSK9 binding.
- Clarify Dose Selection – Justify why 50 µg was chosen for NHPs when higher doses (100–200 µg) showed better responses in mice.
- Discuss Limitations – Address species differences (rodent vs. primate responses) and potential anti-drug antibodies.
- Add a Humanized Mouse Model – Test in hPCSK9 transgenic mice to better predict human responses.
- Evaluate Atherosclerosis Regression – Assess plaque reduction in ApoE⁻/⁻ mice to strengthen CVD relevance.
- Include a Multi-Dose Toxicity Study – Test 6-month chronic dosing in NHPs to confirm long-term safety.
- Assess Synergy with Statins – Since most patients are on statins, test combination therapy.
- Discuss Clinical Trial Design – Propose a Phase I/II study (dosing, endpoints, patient selection).
- Address Cost-Effectiveness – Compare projected pricing vs. mAbs/siRNA (key for global adoption).
- Explore Alternative Adjuvants – Alum/MPL-A may not be optimal; consider TLR agonists for stronger immunity.
- Improve Figure Quality – Current graphs are descriptive but lack mechanistic insights.
- Suggested New Figures:
Mechanism of PCSK9 inhibition by peptide vaccine vs. mAbs/siRNA.
Structural model of peptide-DT conjugate binding to PCSK9.
Proposed clinical development pathway.
- Add a Summary Table – Compare this vaccine with alirocumab, evolocumab, inclisiran, and AT04A.
- Schematic Diagram Suggestions
How the peptide vaccine induces anti-PCSK9 antibodies then blocks LDLR degradation then lowers LDL-C.
Chemical structure, conjugation chemistry (SMPH linker), and adjuvant formulation.
Graph comparing LDL-C reduction in this study vs. mAbs/siRNA in preclinical models.
Round 2
Reviewer 1 Report
Comments and Suggestions for Authors
"Given the close relationship between PCSK9 and NLRP3 inflammasomes (cited and commented on PMID: 36911736, PMID: 40227195), do the authors have any data/hypotheses regarding the use of their novel PCSK9 antipeptide conjugate diphtheria toxoid in this report?"
I don't see any mention of these observations in the text.
Author Response
Response to Reviewers
Reviewer 1:
Comments 1: "Given the close relationship between PCSK9 and NLRP3 inflammasomes (cited and commented on PMID: 36911736, PMID: 40227195), do the authors have any data/hypotheses regarding the use of their novel PCSK9 antipeptide conjugate diphtheria toxoid in this report?"
I don't see any mention of these observations in the text.
Response 1:
We sincerely thank the reviewer for their insightful comment and valuable recommendation. As mentioned in our earlier response, the anti-PCSK9 peptide–diphtheria toxoid (DT) conjugate evaluated in our study has not yet been tested for its specific effects on the NLRP3 inflammasome pathway. However, upon reviewing the referenced literature (PMID: 36911736, PMID: 40227195), we acknowledge the significance of the PCSK9–NLRP3 axis in mediating inflammatory responses. We appreciate the reviewer’s suggestion, as it highlights an important and promising direction for future investigation. In response to this comment, we have now revised introduction section (line no.71 to106) and in the revised manuscript (line no.86 to 106) we have hypothesize that our anti-PCSK9 product may potentially disrupt the PCSK9–NLRP3 inflammasome axis, thereby contributing to both lipid-lowering and anti-inflammatory effects.
We are grateful to the reviewer for helping us improve the quality and depth of the manuscript through this valuable feedback.
Please see the attachment for revised manuscript file

Reviewer 2 Report
Comments and Suggestions for Authors
the authors have made corrections and I now believe I can recommend the manuscript for publication
Author Response
Response to Reviewers
Reviewer 2:
Comment: The authors have made corrections and I now believe I can recommend the manuscript for publication.
Response: We sincerely thank the reviewer for their time, thoughtful feedback, and constructive suggestions throughout the review process. We truly appreciate your positive assessment and recommendation for publication. Your insights have been instrumental in improving the quality and clarity of our manuscript.
Reviewer 3 Report
Comments and Suggestions for Authors
The manuscript is fine and acceptable without further comments
Author Response
Response to Reviewers
Reviewer 3:
Comments: The manuscript is fine and acceptable without further comments.
Response:
We sincerely thank the reviewer for their time and careful evaluation of our manuscript. We truly appreciate your positive feedback and are grateful for your support in accepting the manuscript without further comments.
Round 3
Reviewer 1 Report
Comments and Suggestions for Authors
agree